# Analysis of Factors Affecting the Rate of Latent Tuberculosis Infection and Management in Pediatrics

**DOI:** 10.3390/children9101567

**Published:** 2022-10-17

**Authors:** Hee Won Ma, Hee Soo Lee, Ji Young Ahn

**Affiliations:** 1College of Medicine, Yeungnam University, Daegu 42415, Korea; 2Department of Pediatrics, College of Medicine, Yeungnam University, Daegu 42415, Korea

**Keywords:** latent tuberculosis infection, contact investigation, follow-up, index case, pediatrics

## Abstract

The incidence of tuberculosis remains high in South Korea; the management of latent tuberculosis infection (LTBI) has become the prime target for reducing the infection rate. The management of pediatric LTBI is especially crucial because children can serve as a long-term source of infection upon developing active tuberculosis. Therefore, it is important to assess pediatric LTBI using contact investigation and follow-up. We conducted a retrospective study on children aged between 0 and 18 years who visited our hospital for tuberculosis contact screening from February 2012 to February 2021. Tuberculosis index cases and their clinical characteristics were also reviewed retrospectively. A total of 350 children were investigated, and 68 of 247 (27.5%) were diagnosed with LTBI. The rate of LTBI (r = 7.98, *p* < 0.001) and the risk of loss to follow-up (r = 27.038, *p* < 0.001) were higher in cases with close household contact. Sputum (r = 10.992, *p* < 0.001) and positive acid-fast bacillus (AFB) stain (r = 4.458, *p* = 0.001) in tuberculosis index cases were related to the diagnosis of LTBI in pediatric contacts. Active management is needed for tuberculosis screening in pediatric contacts, especially when the contacts are older and the index case is within the family, and when the index case has sputum and has tested positive for AFB smear.

## 1. Introduction

In diagnosing tuberculosis in children, recent contact with patients having active tuberculosis is prioritized. In particular, many infections arise due to close contact with infected family members [1]. The risk of developing a disease after tuberculosis infection is mainly determined by the age and immune status of the child [2]; the younger the child, the higher the risk of developing active tuberculosis [3]. *Mycobacterium tuberculosis* infection does not immediately progress to active tuberculosis but may exist in the form of a latent tuberculosis infection (LTBI) that is in balance with the immune system [4,5]. Approximately 10% of LTBI cases progress to active tuberculosis [6]. Therefore, diagnosing and treating LTBI is essential for tuberculosis management and as a method of fundamentally eliminating the occurrence of active tuberculosis, which acts as a source of future infection [7]. All children and adolescents with LTBI receive treatment [8] because their likelihood of recent infection is higher than that of adults [8], and the period during which it can progress to tuberculosis in the future and act as an infectious agent is longer [9]. Therefore, when a patient develops active tuberculosis, it is important to investigate for pediatric contacts.

To date, studies on childhood tuberculosis contact in Korea have covered only short research periods (approximately 1 to 3 years), and the number of participants was 60 to 120 in a single-center study [10]. The correlation between the indices of infectious agents and the infection rate was also investigated, but no correlation was found except for in a study at Kangbuk Samsung Hospital in 2017, which announced a correlation with the erythrocyte sedimentation rate (ESR) [10]. Research mainly focuses on the status of contact examination and treatment, such as examination completion rate and treatment. A meta-analysis study was published in The Lancet in 2020, but the analysis focused on the characteristics of the contact rather than the relationship with the source of infection or the characteristics of the source of infection [11].

This study was performed to analyze the infection pattern of children under the age of 18 years following exposure to tuberculosis, the characteristics of the index cases and the pediatric contact, and the difference in the infection rate according to the characteristics of the index cases. Through these analyses, factors related to actual infection were investigated in children exposed to patients with active tuberculosis. In addition, we intended to help prevent and manage tuberculosis in children and adolescents by identifying the status of follow-up management of contacts in cases of tuberculosis exposure.

## 2. Materials and Methods

### 2.1. Study Design

Among the family members or close contacts of patients diagnosed with active tuberculosis, children and adolescents aged 0–18 years who visited Yeungnam University Hospital for tuberculosis contact screening from 1 February 2012, to 28 February 2021, were defined as contacts, and their medical records were investigated retrospectively. We analyzed several characteristics of the pediatric contacts; sex, age, tuberculin skin test (TST) results, relationship with index case, diagnosis of LTBI, and follow-up and/or loss to follow-up. Among the index cases, 85 patients with active tuberculosis were diagnosed at Yeungnam University Hospital. Their medical records were also investigated retrospectively.

A TST was performed on contacts. For the TST, 0.1 mL of 2TU PPD RT 23 (TU, tuberculin unit; PPD, purified protein derivative; RT 23; Statens Serum Institut, Copenhagen, Denmark) was injected intradermally into the skin on the inside of the arm 10 cm below the elbow to produce a 6–10 mm diameter wheal. Thereafter, the longest diameter of the induration formed after 48 to 72 h was palpated with the fingertip [8].

The screening and treatment of pediatric contacts were conducted in accordance with the latest tuberculosis treatment guidelines according to the diagnosis period of the contact [8,12,13]. The guidelines revised in 2011 were applied in the cases diagnosed until July 2014; the guidelines revised in 2014 were applied in the cases diagnosed until April 2017; and the guidelines revised in 2020 were applied in the cases of subsequent diagnosis.

According to the 2011 guidelines, children under the age of 5 years were tested with a TST, and if active tuberculosis was excluded for >10 mm, LTBI treatment was initiated; if it was <10 mm, re-testing with the TST was performed after 8 weeks. If there was an increase of ≥6 mm in the re-test, treatment for LTBI was initiated; otherwise, isoniazid treatment was stopped. In children and adolescents aged ≥5 years, if the TST result was <10 mm, unlike in children under the age of 5 years, the TST was re-executed 8 weeks later without isoniazid treatment [12].

According to the 2014 guidelines, children under the age of 2 years were tested for TST, and if active tuberculosis was excluded for ≥10 mm, LTBI treatment was initiated; and if it was <10 mm, re-testing with TST was performed after 8 weeks. If there was an increase of ≥6 mm in the re-test, treatment for LTBI was initiated; otherwise, isoniazid treatment was stopped. In children and adolescents aged ≥2 years, if the TST result was <10 mm, unlike in children under the age of 2 years, the TST was re-executed 8 weeks later without isoniazid treatment [13].

For the medical records of 85 index cases that could be identified from the contact records, the presence or absence of symptoms (fever, cough, and sputum), imaging findings, sputum test, bronchopulmonary lavage test (acid-fast bacilli smear, culture, and polymerase chain reaction test), leukocyte count, and ESR were investigated to analyze the association with the latent tuberculosis diagnosis of the contact.

This study was approved by the Ethics Committee of Yeungnam University Hospital (YUMC 2022-08-043). The need for informed consent was waived by the committee.

### 2.2. Statistical Analysis

Statistical Package for the Social Sciences (SPSS) Ver. 25 (SPSS Inc., Chicago, IL, USA) was used for data analysis. A cross-analysis of the chi-squared test and Fisher’s exact test was performed to compare the characteristics of the contact according to their relationship with the index cases. The Mann–Whitney test was performed after cross-analysis and a normality test for the analysis of the contact examination results according to the characteristics of the contact. For the correlation between the characteristics of the index cases and the latent tuberculosis diagnosis of the contact, cross-analysis of the chi-squared test and Fisher’s exact test was performed when the characteristics of the index cases were categorical data, and in the case of continuous data, the Kruskal–Wallis test was performed after the normality test. Logistic regression was performed to identify factors affecting the diagnosis and follow-up of the contacts. If the *p* value was <0.05, it was judged to be statistically significant.

## 3. Results

### 3.1. Participant Demographics

The study population included a total of 350 pediatric contacts (200 males and 150 females). To compare age groups, age was divided into 0–6, 7–12, and 13–18 years, based on Korean school years. There were 206 contacts in the 0–6 years, 79 contacts in the 7–12 years, and 65 contacts in the 13–18 years age groups, respectively. Excluding patients with loss to follow-up, 68 contacts out of 247 (27.5%) were diagnosed with LTBI, 26 of whom were diagnosed upon follow-up. Twelve children had contact with isoniazid-resistant index cases. A total of 106 contacts were lost to follow-up during the diagnosis and treatment of LTBI. There were no HIV patients of the contacts.

Contacts were divided into five groups: 0—grandparent, 1—parent, 2—sibling, 3—relative, and 4—community, according to their relationship with the index cases. Age, sex, and tuberculosis contact screening results were arranged accordingly (Table 1). There were 87 children in Group 0, 163 in Group 1, 14 in Group 2, 8 in Group 3, and 78 in Group 4. In Group 4, 69 cases contacted a specific staff member who worked in the neonatal intensive care unit during the same period. Group 2 had the highest mean age, and Group 4 had the lowest. The distribution of age groups according to contact history showed a statistically significant difference (*p* < 0.001). As for sex, there was no statistical difference in distribution according to contact history (*p* = 0.342).

### 3.2. Correlations between Contact History and Tuberculosis Screening Test Results

The LTBI diagnosis rate was 42% (21 out of 50) in Group 0, 37.1% (39 out of 105) in Group 1, 25% (2 out of 8) in Group 2, 37.5% (3 out of 8) in Group 3, and 3.8% (3 out of 76) in Group 4 (Figure 1). The distribution of LTBI between the groups was significantly different (*p* < 0.001), and Group 4 had a lower LTBI rate than Group 0 or Group 1.

Forty-three percent (37 of 86) were lost to follow-up in Group 0, 37% (60 out of 162) in Group 1, 42.9% (6 out of 14) in Group 2, none in Group 3, and 2.6% (2 out of 78) in Group 4. The distribution of contacts who were lost to follow-up was also significantly different between groups (*p* < 0.001), and Group 4 had the smallest number lost.

Tuberculosis screening test results were analyzed by classifying contacts with grandparents, parents, and siblings as index cases as close household contacts, and contacts with relatives and community index cases as close non-household contacts. Patients with close household contacts had a higher rate of LTBI (r = 7.98, *p* < 0.001) and more loss to follow up throughout the whole screening and treatment procedure (r = 27.038, *p* < 0.001) compared with close non-household contacts. There was no difference between the two groups in the rate of diagnosis of LTBI at the first TST and at follow-up TST (*p* = 0.668). In addition, there were more cases of isoniazid-resistant index cases in close household contacts than in close non-household contacts (*p* = 0.011). There was no statistically significant difference in the initial TST result (*p* = 0.142), changes in TST result (*p* = 0.861), or loss to follow-up during the treatment procedure between close household and close non-household contacts (*p* = 0.617).

### 3.3. Correlation between Age and Tuberculosis Screening Test Results

Among the contacts, members of the group who tested positive for LTBI were significantly older than those of the group who tested negative for LTBI (*p* < 0.001) (Figure 2). Among the contacts, the mean age of children was significantly higher in the loss to follow-up group than in the follow-up group (*p* < 0.001) (Figure 2). There was no statistically significant difference according to age in the initial TST results (*p* = 0.100) or TST result changes (*p* = 0.257).

### 3.4. Risk Factors of LTBI and Loss to Follow-Up

Among the characteristics of contacts, age and contact type were found to affect LTBI diagnosis and follow-up loss. Compared with the 0–6 years age group, the 7–12 years age group had a higher risk of LTBI (odds ratio [OR], 2.125; 95% confidence interval [CI], 1.053–4.289), and the risk of LTBI was also higher in the 13–18 years age group (OR, 2.528; 95% CI, 1.137–5.622) (Figure 3). When both age group and contact type were considered as risk factors for LTBI diagnosis, close household contacts had a higher risk than close non-household contacts (OR, 7.397; 95% CI, 2.918–18.751), but age group did not have statistically significant effects (Figure 3).

As for loss to follow-up, significant results were found both when age was considered as a single factor and when contact history was concurrently considered. When age was taken as a single factor, compared with the 0–6 years age group, the 7–12 years (OR, 3.529; 95% CI, 1.993–6.252) and 13–18 years age group (OR, 5.010; 95% CI, 2.741–9.154) both had an increased risk of loss to follow-up (Figure 3). When the age group and contact type were both taken as risk factors, the risk of loss to follow-up increased more in the case of close household contacts than in the close non-household contacts (OR, 17.492; 95% CI, 4.097–74.676) (Figure 3), and both the 7–12 years (OR, 1.953; 95% CI, 1.077–3.542) and 13–18 years age group (OR, 2.795; 95% CI, 1.494–5.228) showed an increased risk of loss to follow-up compared with the 0-6 years age group.

### 3.5. Clinical Characteristics of Index Cases

A total of 84 index cases were identified, of which 42 were male and 42 were female. Of these index cases, 54 (74.0%) had one or more symptoms of tuberculosis, 43 had a cough (58.9%), 33 had sputum (51.6%), and 11 had fever (17.7%). Sputum, bronchoalveolar lavage, lung image findings, and laboratory results are described in Table 2.

### 3.6. Comparisons of Index Cases between LTBI-Positive and LTBI-Negative Participants

Among the contacts, 68 children and adolescents were diagnosed with LTBI, and 179 children and adolescents ultimately tested negative. Among the symptoms of tuberculosis, sputum showed a significantly higher frequency in the index cases of the LTBI-positive group than in the LTBI-negative group (r = 10.992, *p* < 0.001). There was no statistically significant difference in the presence of cough or fever symptoms (Table 3).

Regardless of the sample collection method, positive findings of the acid-fast bacilli (AFB) smear (r = 4.458, *p* = 0.001) of the index cases were related to the diagnosis of LTBI in pediatric contacts. Positive bronchoalveolar lavage AFB smear was significantly related to the diagnosis of LTBI in the pediatric contacts (r = 4.831, *p* = 0.005), while positive sputum AFB smear was not (Table 3).

Mean blood ESR level, lymphocyte ratio, and neutrophil–lymphocyte ratio were higher in the LTBI-positive group than in the LTBI-negative group, but there was no statistically significant difference.

## 4. Discussion

Tuberculosis is transmitted through the respiratory tract during close contact with an individual with active tuberculosis. Once transmitted, the bacteria exist in the form of asymptomatic LTBI, and around 10% of infected patients develop active, symptomatic tuberculosis throughout their lifetime [14]. Tuberculosis is a common disease, with 10 million cases worldwide, 130 cases per 100,000 individuals, and approximately 1.5 million tuberculosis-related deaths per year [15]. Korea has the highest tuberculosis prevalence and the third-highest mortality rate among Organization for Economic Co-operation and Development country members [16].

The incidence of tuberculosis in Korea has been on a steady decline since 2011 [16], and the National Tuberculosis Elimination Project with private–public cooperation (implemented in 2011), along with the improvement of healthcare quality and the development of public health, have attributed to the decline [17,18]. In the tuberculosis management project, the most important factor in reducing transmission is to check and treat LTBI by investigating the contacts of patients with active tuberculosis. The contact screening of children and adolescents is crucial because it can help treat LTBI and prevent the occurrence of active tuberculosis, thereby reducing the possibility that children and adolescents with LTBI will transmit tuberculosis when they become adults in the future [2]. Therefore, the thorough management of all child and adolescent contacts is required. However, to date, there are insufficient domestic studies on the relationship between the characteristics of the index cases and LTBI rate, and the management of child and adolescent contacts.

In this study, we aimed to analyze the pediatric tuberculosis infection pattern after exposure, the characteristics of index patients and contacts, and the difference in LTBI incidence according to the index patients’ features. The purpose of this study was to identify factors related to infection, follow-up care, and the treatment status of contacts.

In this study, the rate of LTBI is similar to the contact infection rate of 28.1% in high-income countries, according to the meta-analysis of Fox et al. [19], and significantly lower than the pediatric contact LTBI rate of low- and median-income countries, which was 40% [20].

In this study, the rate of diagnosis of LTBI in the close household contact group was significantly higher. This is consistent with the results of previous studies which showed that the risk of tuberculosis infection is higher with close family contacts [2,21], and that the more the geographical proximity to an infected person and the higher the activities performed together, the higher the infection rate [21].

Loss to follow-up was also significantly higher for close household contacts than for close non-household contacts. However, there was no difference between the two groups in the rate of LTBI-positive participants who were diagnosed on the first TST or follow-up test. That is, individuals with close household contact were more likely to be diagnosed with LTBI than those with close non-household contacts, and even though there was no difference in the rate of diagnosis among the first and follow-up screening tests, it was difficult to identify potential LTBI diagnosis due to loss to follow-up. According to the results of the nationwide screening project for close household contacts of loss to follow-up, the risk of tuberculosis in contacts who tested negative for LTBI with a single test and with consecutive tests, compared with contacts who did not undergo screening tests, were 0.36 and 0.09, respectively [22]. In other words, the tuberculosis incidence in those who underwent screening was significantly lower, and even lower when all consecutive tests were completed rather than just a single test. Combined with the results of this study, in order to identify patients with LTBI and lower the incidence of active tuberculosis in the future, it is necessary to strengthen the monitoring of close household contacts during the contact screening process and increase the rate of completion for follow-up tests.

There were also differences in diagnosis and loss to follow-up between the age groups. In particular, when contact history was also considered as a risk factor, the risk of loss to follow-up was higher with close household contact and older age groups. This was similar to the domestic study results which showed that the younger the age of the contact, the higher the LTBI screening completion rate [12,13]. The LTBI diagnosis rate was also higher in the older age group, which agrees with previous studies showing that the older the age, the higher the LTBI diagnosis rate [11,23,24,25,26]. The 2018 World Health Organization report showed that tuberculosis exposure increased with age [26]. Therefore, it is necessary to thoroughly manage contact screening for school-age children and adolescents with index cases within the family.

The characteristics of index cases that affected the diagnosis of contact LTBI were sputum and positive AFB smear. Although there were various results on which the characteristics of index cases were risk factors for the transmission of tuberculosis, positive sputum AFB smear was classified as a risk factor for transmission in many studies [2,22,27,28,29,30]. Another risk factor among the symptoms of index cases was cough [28], and among the imaging findings risk factors included the extent of the lung field invaded on chest X-ray findings [2], cavity [22], and left lung right lobe lesion [28].

There are some limitations to this study. First, this study was conducted in a single center, and thus, it was difficult to interpret the results of the study as a nationwide trend due to the regional bias in study participants, such as one index case causing 69 contacts. Second, as this was a retrospective study, it was difficult to identify all index cases, the presence of symptoms, and the use of therapeutic agents of the index cases. In addition, although the relationships between index cases and corresponding contacts were identified, the exact contact duration was unknown. Third, as the research was conducted to find the index cases through contact screening records, 134 index cases were unknown; therefore, the data on these index cases were incomplete. Fourth, since most index cases identified were close household contacts, there could be selection bias in the data of index cases.

Nevertheless, this study may be the first report on both the index cases and contacts. In addition, it was possible to find risk factors for tuberculosis transmission in index cases and examine the status of contact screening for tuberculosis prevention. Since this study was based on data of child contacts with tuberculosis screenings conducted from 2012–2021, the number of pediatric contacts was considerable even for a single-center study and could be helpful for future studies despite the limitations listed above.

## 5. Conclusions

In conclusion, when screening for tuberculosis in pediatric contacts, more active management is needed in the overall contact screening process when the contacts are older and the index case is within the family, and when the index case has sputum and has tested positive for AFB smear.

## Figures and Tables

**Figure 1 children-09-01567-f001:**
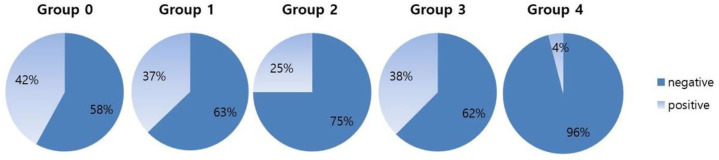
Latent tuberculosis diagnosis rate in each group. Group 0, grandparent; Group 1, parent; Group 2, sibling; Group 3, relative; Group 4; community.

**Figure 2 children-09-01567-f002:**
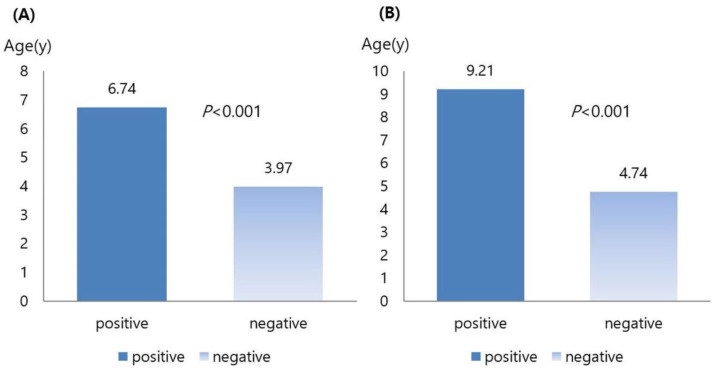
(**A**) Comparison of age between positive and negative diagnosis of latent tuberculosis (**B**) Comparison of age between positive and negative loss to follow-up.

**Figure 3 children-09-01567-f003:**
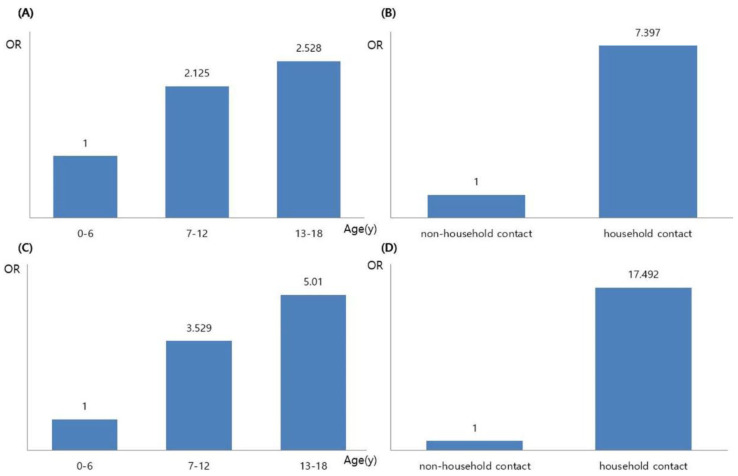
Risk factors of latent tuberculosis infection (LTBI) and loss to follow-up. (**A**) Comparison of age affecting LTBI; (**B**) Comparison of contact type affecting LTBI; (**C**) Comparison of age affecting loss to follow-up; (**D**) Comparison of contact type affecting loss to follow-up. OR, odds ratio.

**Table 1 children-09-01567-t001:** Participant demographics.

Characteristics	Relationship between the Index Case and the Contact					Total (*n*)	*p* Value
	Grandparent	Parent	Sibling	Relative	Community		
Age group (y)							<0.001
0–6	44 (50.6)	78 (47.9)	0 (0.0)	7 (87.5)	77 (98.7)	206	
7–12	25 (28.7)	51 (31.3)	2 (14.3)	1 (12.5)	0 (0.0)	79	
13–18	18 (20.7)	34 (20.9)	12 (85.7)	0 (0.0)	1 (1.3)	65	
Sex							0.342
Male	54 (62.1)	94 (57.7)	8 (57.1)	2 (25)	42 (53.8)	200	
Female	33 (37.9)	69 (42.3)	6 (42.9)	6 (75)	36 (46.2)	150	
Latent tuberculosis infection							
Negative	29 (58)	66 (62.9)	6 (75)	5 (62.5)	73 (96.1)	179	
Positive	21 (42)	39 (37.1)	2 (25)	3 (37.5)	3 (3.9)	68	<0.001
Isoniazid-resistant							
Negative	37 (90.2)	104 (92.9)	5 (100)	3 (100)	72 (100)	221	
Positive	4 (9.8)	8 (7.1)	0 (0.0)	0 (0.0)	0 (0.0)	12	0.011
Loss to Follow-up							
No	49 (56.3)	103 (63.2)	8 (57.1)	8 (100)	76 (97.4)	244	
Yes	38 (43.7)	60 (36.8)	6 (42.9)	0 (0.0)	2 (2.6)	106	<0.001

Values are presented as number (%).

**Table 2 children-09-01567-t002:** Clinical characteristics of the index cases.

Characteristics	Number of Index Cases
Sex	
Male	42 (50)
Female	42 (50)
Symptoms	
Cough	43 (58.9)
Sputum	33 (51.6)
Fever	11 (17.7)
All symptoms	54 (74.0)
Tuberculosis sputum test	
Positive AFB stain	22 (37.9)
Positive PCR test	17 (45.9)
Positive culture	26 (56.5)
Tuberculosis BAL test	
Positive AFB stain	12 (21.1)
Positive PCR test	36 (62.1)
Positive culture	30 (51.7)
Drug-resistant tuberculosis	
Isoniazid-resistant	6 (7.6)
Chest radiographic findings	
Cavity	19 (22.6)
Tree-in-bud	22 (26.2)
Bronchogenic spread	20 (23.8)
Consolidation	31 (36.9)
Nodular infiltration	66 (78.6)
Pleural effusion	11 (13.1)
Laboratory findings	
ESR (mm/h)	39.32 ± 28.03
WBC (/μL)	7306.38 ± 2971.39
Neutrophil ratio (%)	63.71 ± 10.95
ANC (/uL)	4859.82 ± 2722.49
Lymphocyte ratio (%)	25.09 ± 9.31
ALC (/uL)	1666.45 ± 594.10
Neutrophil–lymphocyte ratio	3.32 ± 2.62
Platelet (/uL)	274014.49 ± 83424.23

Values are presented as mean ± standard deviation or number (%). AFB, acid-fast bacilli; PCR, polymerase chain reaction; BAL, bronchoalveolar lavage; WBC, white blood cell; ANC, absolute neutrophil count; ALC, absolute lymphocyte count; ESR, erythrocyte sedimentation rate.

**Table 3 children-09-01567-t003:** Comparison of index case features between children with and without latent tuberculosis infection.

Characteristics	R	*p* Value
Symptom		
Cough	0.406	0.065
Sputum	10.992	<0.001
Fever	0.851	1
All symptoms	0.935	1
Tuberculosis diagnostic test		
Positive AFB stain	4.458	0.001
Positive BAL AFB stain	4.831	0.005
Positive sputum AFB	0.724	0.592
Positive PCR test	0.179	<0.001
Positive BAL PCR test	0.186	0.004
Positive sputum PCR test	0.6	0.53
Positive culture	0.418	0.059
Positive BAL culture	0.282	0.018
Positive sputum culture	0.756	0.771

AFB, acid-fast bacilli; BAL, bronchoalveolar lavage; PCR, polymerase chain reaction.

## Data Availability

The data presented in this study are available on request from the corresponding author. The data are not publicly available due to ethical reasons.

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
