# Peer review of "Analysis of Factors Affecting the Rate of Latent Tuberculosis Infection and Management in Pediatrics"

_children, 2022, doi:10.3390/children9101567_

Round 1
Reviewer 1 Report
Dear Authors:
I recognize the meritis of yor manuscript. The number of cases is relevant and time frame long. However, there is insufficient novelty to be considered a major advance in this field (TB contact tracing in children).
In Results there is no information about HIV status of the contacts. I suggest to add this, because the National Protocol for TB probably consider TST ≥ 5mm as positive (not 10 mm).
DIscussion should be reduced and avoid to repeat Results. I suggest to review Conclusion, as well. " Additional large-scale studies are needed to understand the complex relationship .... " are not Conclusions in this article. .
Author Response
1. In Results there is no information about HIV status of the contacts. I suggest to add this, because the National Protocol for TB probably consider TST ≥ 5mm as positive (not 10 mm).
Thank you for your comment. ‘There were no HIV patients of the contacts.’ We added it (line 129).
2. Discussion should be reduced and avoid to repeat Results. I suggest to review Conclusion, as well. " Additional large-scale studies are needed to understand the complex relationship .... " are not Conclusions in this article.
Thank you for your comment.
We reduced some sentences in Discussion (line 256, 260, 284). That sentences are the same as the results.
We removed ‘Additional large-scale studies are needed to understand the complex relationship .... ‘.
Reviewer 2 Report
Reviewer’s Comment
The article by Ma et al on “Analysis of factors affecting the rate of latent tuberculosis infection and management in pediatrics” discuss about the analysis of infection pattern of children under the age of 18 years following exposure to tuberculosis, the characteristics of the index cases and the pediatric contact, and the difference in the infection rate according to the characteristics of the index cases. They further showed that the rate of LTBI (r=7.98, P<0.001) and risk of loss to follow-up (r=27.038, 18 P<0.001) were higher in cases with close household contact. Sputum (r=10.992, P<0.001) and positive 19 acid-fast bacillus stain (r=4.458, P=0.001) in tuberculosis index cases were related to contact with 20 diagnosed LTBI cases. The paper is well written, however, it needs to address few points before consideration for acceptance.
Comments
1. It should be clarified whether these 85 index cases were contact earlier that developed TB later? The manuscript writing styles does not clarifies it and readers may confuse whether we selected 85 cases who were TB positive by retrospective analysis or they were contact of earlier TB patients and then developed TB subsequently and for this study they are index cases.
2. Line 72 to 75 does not clarifies whether TST test was performed on contacts (Children) or patients?
3. The table 1 needs to be redrawn. It is very difficult to understand the tables as legends are missing? For example in age row authors have mentioned 7.5, 7.4, etc. However, age is 0-6, 7-12? It will make the readers confused?
4. Further, in table bracket, I presume that authors have mentioned percentage. But no legend is there to denote it.
5. There is some contradiction between table and test. For example, in result section 3.2, the LTB diagnosis rate mentioned 42% in group 0. However, from table1, one interpretation is 30.9%. The situation is similar with other data also. Authors need to clarify it.
6. The same issue is with lost to follow up. There seems to be mismatch between table and text.
7. Further, author can present the data of LTB diagnosis as pie chart for different group. It will help to understand the result in a better way and with clarity.
8. In result section 3.3, merely writing the text in the form of result is creating confusion. The author can present the data as bar graph, showing correlation between age and LTB positivity.
9. Same comments can be applied for result section 3.4. The author can present the data as bar graph.
10. Table 2 is missing.
Author Response
- It should be clarified whether these 85 index cases were contact earlier that developed TB later? The manuscript writing styles does not clarify it and readers may confuse whether we selected 85 cases who were TB positive by retrospective analysis, or they were contact of earlier TB patients and then developed TB subsequently and for this study they are index cases.
Thank you for your comment. We removed ‘were retrospectively identified through the medical records and used for analysis of characteristics’ and added ‘Their medical records were also investigated retrospectively.’ (line 73)
- Line 72 to 75 does not clarify whether TST test was performed on contacts (Children) or patients?
Thank you for your comment. We added ‘TST was performed on contacts’(line 75).
- The table 1 needs to be redrawn. It is very difficult to understand the tables as legends are missing? For example in age row authors have mentioned 7.5, 7.4, etc. However, age is 0-6, 7-12? It will make the readers confused?
Thank you for your comment. We removed mean±standard deviation and all of values are presented number (%).
- Further, in table bracket, I presume that authors have mentioned percentage. But no legend is there to denote it.
Thank you for your comment. We described a legend below the Table (line 140).
- There is some contradiction between table and test. For example, in result section 3.2, the LTB diagnosis rate mentioned 42% in group 0. However, from table1, one interpretation is 30.9%. The situation is similar with other data also. Authors need to clarify it.
Thank you for your comment. The percentage in Table 1 means when contacts were divided into five groups. All of numbers in five groups are 100 percentage. In section 3.2, percentage means positive and/or negative numbers in each group. We corrected the percentage in Table 1 as same as section 3.2.
- The same issue is with lost to follow up. There seems to be mismatch between table and text.
Thank you for your comment. The percentage in Table 1 means when contacts were divided into five groups. All of numbers in five groups are 100 percentage. In section 3.2, percentage means positive and/or negative numbers in each group. We corrected the percentage in Table 1 as same as section 3.2.
- Further, author can present the data of LTB diagnosis as pie chart for different group. It will help to understand the result in a better way and with clarity.
Thank you for your comment. We added pie chart for different group (Fig.1).
- In result section 3.3, merely writing the text in the form of result is creating confusion. The author can present the data as bar graph, showing correlation between age and LTB positivity.
Thank you for your comment. We added bar graph (Fig.2).
- Same comments can be applied for result section 3.4. The author can present the data as bar graph.
Thank you for your comment. We added bar graph.(Fig.3)
- Table 2 is missing.
Thank you for your comment. We added Table 2.
Round 2
Reviewer 1 Report
Congratulations for this important paper. I recognize the local interest of your data and for TB endemic areas, as well.